# Phylogeography as a Proxy for Population Connectivity for Spatial Modeling of Foot-and-Mouth Disease Outbreaks in Vietnam

**DOI:** 10.3390/v15020388

**Published:** 2023-01-29

**Authors:** Umanga Gunasekara, Miranda R. Bertram, Nguyen Van Long, Phan Quang Minh, Vo Dinh Chuong, Andres Perez, Jonathan Arzt, Kimberly VanderWaal

**Affiliations:** 1Veterinary Population Medicine, University of Minnesota, St. Paul, MN 55108, USA; 2Foreign Animal Disease Research Unit, USDA-ARS, Plum Island Animal Disease Center, Southold, NY 11957, USA; 3Department of Animal Health, Ministry of Agriculture and Rural Development, Hanoi, Vietnam

**Keywords:** Bayesian space-time models, INLA, outbreak risk, mobility, host movement, phylodynamics, foot-and-mouth disease virus, FMD, Vietnam

## Abstract

Bayesian space–time regression models are helpful tools to describe and predict the distribution of infectious disease outbreaks and to delineate high-risk areas for disease control. In these models, structured and unstructured spatial and temporal effects account for various forms of non-independence amongst case counts across spatial units. Structured spatial effects capture correlations in case counts amongst neighboring provinces arising from shared risk factors or population connectivity. For highly mobile populations, spatial adjacency is an imperfect measure of connectivity due to long-distance movement, but we often lack data on host movements. Phylogeographic models inferring routes of viral dissemination across a region could serve as a proxy for patterns of population connectivity. The objective of this study was to investigate whether the effects of population connectivity in space–time regressions of case counts were better captured by spatial adjacency or by inferences from phylogeographic analyses. To compare these two approaches, we used foot-and-mouth disease virus (FMDV) outbreak data from across Vietnam as an example. We identified that accounting for virus movement through phylogeographic analysis serves as a better proxy for population connectivity than spatial adjacency in spatial–temporal risk models. This approach may contribute to design surveillance activities in countries lacking movement data.

## 1. Introduction

Space–time risk models are often applied to outbreak data to understand patterns and drivers of pathogen spread. In Bayesian space–time models, risk of disease is considered as a spatial process whereby spatial correlations in case counts are captured by accounting for the contiguity or adjacency of spatial units (i.e., states, provinces, etc.) [1,2]. Such space–time regressions have been widely used in disease mapping studies to identify high-risk areas for both animal and human diseases [3,4,5]. However, spatial adjacency is often an imperfect proxy for population connectivity, especially for humans, migratory wildlife, and highly mobile populations of livestock that may frequently be transported long distances. Unfortunately, mobility data is rarely available for many host–pathogen systems. For example, while livestock movement data is a critical component of outbreak response for foot-and-mouth disease virus (FMDV) incursions in FMD-free countries, such data is usually not available for control efforts in countries where FMDV is endemic. However, patterns of viral dissemination (inferred from phylogeographic models) could serve as a proxy for the underlying connectivity of the host population [6,7].

FMD is a highly contagious disease that is endemic in Southeast Asia (SEA) affecting pigs, cattle, buffaloes, and small ruminants. The disease is caused by an *Aphthovirus* in the *Picornaviridae* family, and serotypes O, A, and Asia 1 have been identified in the region [8,9]. In endemic settings, clinical signs of the disease such as vesicles and pyrexia cause production losses to farmers. FMD is challenging to control in SEA in part due to diverse animal husbandry practices in the growing economies of the region. In addition, the occurrence of FMD outbreaks in SEA is spatially and temporally variable, and some countries are free from FMD [10]. Lack of measures to track animal movements within and between countries and undocumented livestock markets are major obstacles to disease preparedness and hinder investigations of patterns of disease occurrence of contagious livestock diseases in the region, including in Vietnam [10,11]. Additionally, the potential epidemiologic role of carriers and neoteric subclinical infections remains poorly understood [12].

Phylogeography has been used to identify patterns of FMDV transmission among countries and regions, as demonstrated by the spread of lineages O/ME-SA/PanAsia2 and O/ME-SA/Ind-2001e [10]. O/ME-SA/Ind-2001e was first identified in SEA during 2013–2017, with phylogeographic patterns demonstrating the movement of the virus from India to Vietnam [9,13,14]. Similar patterns were identified for O/ME-SA/PanAsia during 2010–2014. Within Vietnam, phylogeographic analysis demonstrated frequent dissemination of O/ME-SA/PanAsia from the South-Central Coast and Northeast to other parts of the country, presumably due to livestock movement that occurs from central areas of the country to the north and south [15]. Taken together, these studies highlight the transboundary nature of FMDV circulation in SEA, and patterns of spread within Vietnam. These phylogeographic-based inferences of viral movements might be used as a proxy for host population connectivity.

For space–time risk models of case counts, we hypothesized that historical patterns of viral movement are a better criterion to define population connectedness between spatial units compared to spatial adjacency. Using Vietnam as an example, we constructed discrete trait phylogeographic models for FMDV serotype O in Vietnam and SEA. We then used the inferred transition rates between each province as the connectivity matrix in a Bayesian space–time regression and compared this model’s ability to explain spatiotemporal variability in the relative risk of outbreaks with conventional approaches based on spatial adjacency. We also used these models to delineate high-risk areas for FMD in Vietnam and identify spatial risk factors for FMD outbreaks, such as livestock density, international borders, and phylogeographically inferred introductions of FMDV from other countries.

## 2. Materials and Methods

### 2.1. Population Description

The livestock market system in SEA is complex. Both livestock and meat products are transported across borders following a system where supply meets the highest demand. There is evidence that livestock move across Vietnam to Thailand via surrounding countries [16] and more recent documentation of livestock movement towards China from Lao PDR, Thailand, and Vietnam [17]. Vietnam is a hub for animal movements in the SEA region and cannot be considered as a separate entity.

The country is divided into 63 provinces grouped in eight major agriculture zones, referred to as Northwest, Northeast, Red River Delta, North Central Coast, South-Central Coast, Central Highlands, Southeast, and Mekong River Delta. Vietnam has had an FMD control program in place since 2006 (National Program for Prevention and Control of Foot-and-Mouth Disease), and at present, Vietnam is in stage 3 of the OIE/FAO progressive control pathway (PCP). Biannual vaccination of cattle and buffalo is conducted free-of-charge in border provinces and for a fee in other areas of the country [18]. Most (about 85%) of livestock farms in Vietnam are small-scale farms [19]. Pig production supplies 77% of the total meat production in Vietnam. The pig production system has been moving from small-scale backyard systems to large-scale commercial farming systems [20]. There is evidence that pigs from the small-scale farmers move across different agriculture zones for slaughter and production purposes, and especially the zones in the northern part of the country are highly connected for trading purposes [21]. Cattle and buffalo production stand at 18% and 4%, respectively. However, cattle and buffalo farms are more evenly distributed across the country compared to the pig farms concentrated in the northern part of the country (FAOSTAT 2020). Most cattle and buffaloes are reared under extensive management systems for various purposes such as trade, meat production, and draught [16,20].

### 2.2. Data Overview

In this study, we utilized two types of data available from Vietnam: (a) data on the reported number of FMD cases per province per year (Department of Animal Health, Vietnam), and (b) FMDV VP1 sequences generated from various research and surveillance projects conducted in Vietnam [8,9,15,22,23,24,25].

Reported numbers of clinically-infected cattle, buffalo, pigs, and goats, along with estimated outbreak dates, were available for 2007–2017 from the Department of Animal Health (DAH), Vietnam. Data had been collected via passive surveillance at the commune level on a daily/weekly basis by the local sub-Department of Animal Health [18]. Most cases were clinically diagnosed, whereas some cases were laboratory-confirmed. A case was defined as an animal diagnosed as infected by local animal health professionals by the clinical signs only and an outbreak was defined as a group of epidemiologically linked cases (WAOH, Terrestrial code) during the considered time period. For cattle, 49,306 cases were reported from 1677 outbreaks (2007–2017) with a mean of 29 (SD 59) infected animals per outbreak. For buffalo, 70,118 cases were reported from 1841 outbreaks (2007–2017) with a mean of 38 (SD 69) infected animals per outbreak. Province-level livestock population data (cattle, buffalo, pigs) for the year 2018 were also available from the General Statistics Office of Vietnam. These data were used to calculate standardized incidence ratios, and to construct space–time regressions (see details below). A polygon shapefile of Vietnam’s provinces was used to generate a 0/1 matrix that summarizes which provinces are spatially adjacent (hereafter, referred to as the spatial adjacency matrix).

In total, 306 FMDV serotype O VP1 sequences were available from 53 provinces in Vietnam, representing all eight agriculture zones. Not enough FMDV whole genome sequences were available to perform this analysis from a whole genome perspective. Most (*n* = 193, 63%) sequences were collected from farms and slaughterhouses as part of active surveillance of clinical and sub-clinical bovines conducted by our collaborative team (see below for details), whereas the remaining 113 sequences (36%) were downloaded from GenBank (https://www.ncbi.nlm.nih.gov/genbank/). All Vietnam sequences used in the analysis included meta-data pertaining to location (to the state-level) and date. After removing duplicate sequences (from groups of 100% identical sequences), a total of 267 sequences from Vietnam were used for the analyses. Most of those sequences (*n* = 132, 49%) were from cattle, whereas 80 sequences were from pigs, 28 sequences were from buffaloes, and in the remaining 27 (10%) data on the host species were not available. All available sequences with date information from adjacent countries during the period of 2000–2017 were also obtained from GenBank, including sequences from Thailand (*n* = 41), Malaysia (*n* = 37), Laos (*n* = 32), Cambodia (*n* = 4), and China (*n* = 19). The data associated with those 400 sequences (267 from Vietnam and 133 from neighboring countries) were then used in Bayesian phylogeographic models (see details below) to infer rates of viral movement between different agricultural zones, which were then used to generate a 0/1 matrix that summarizes which provinces showed evidence of population connectivity based on patterns of historical viral dispersal (hereafter, referred to as the phylogeographic matrix).

### 2.3. Sequence Acquisition

Oropharyngeal fluid (collected by our team from subclinical animals at farms and slaughterhouses) and epithelium (outbreak) samples were screened for the presence of FMDV using virus isolation (VI), followed by detection of viral RNA in VI supernatant using qRT-PCR as previously described [12,26]. Samples that were positive for viral RNA were subjected to sequencing using one of several methods as previously described [25]. Briefly, samples from 2013–2015 were sequenced using the Sanger method to obtain VP1 sequences [8], or by next generation sequencing (NGS) of overlapping RT-PCR amplicons covering the full ORF [15]. Samples from 2016–2017 were sequenced by NGS of RT-PCR amplicons covering the P1 region [27]. Finally, sequences from 2018–2019 were sequenced by NGS using random and FMDV-specific primers to obtain the complete genome as previously described [28,29]. All NGS sequencing was performed using the Illumina NextSeq platform. Read quality filtering, de novo assembly, and assembly to previously published references of regionally endemic lineages were implemented in CLC Genomics Workbench v12 (Qiagen). Sequences of the VP1 region were utilized in this study, as this region was the segment available consistently across years and in GenBank.

### 2.4. Phylodynamic Analyses

Sequences from Vietnam and neighboring countries were aligned against the reference strain O/LAO/2/2006 (representing O/ME-SA/PanAsia) using Muscle in MEGA-X [30]. Sequences were checked for recombination using RDP4 [31], and no recombinants were detected. Initial maximum likelihood trees revealed three distinct clades corresponding to lineages O/SEA/Mya-98 (146 sequences), O/ME-SA/PanAsia (229 sequences), and O/Cathay (25 sequences).

Discrete-space phylogeographic analyses were performed for O/ME-SA/Pan Asia and O/SEA/Mya-98 separately and for all sequences combined using Bayesian Evolutionary Analysis by Sampling Trees (BEAST v1.10.1.2) [32]. A lineage-specific model was not constructed for O/Cathay due to insufficient data. For these analyses, each agriculture zone within Vietnam (eight zones) and each neighboring country (four countries) were used as discrete traits. Agricultural zones rather than provinces were used within Vietnam to achieve adequate numbers of sequences per location for the analysis due to limited sequence data availability from each province of Vietnam. Because our intent for this analysis was to identify patterns of FMDV movement within and between countries, all available sequences regardless of host species were included in this analysis.

Maximum likelihood trees for each lineage and the combined data were analyzed with Tempest (v1.5.3) [33] to establish whether the phylogenies had sufficient temporal signals for subsequent analyses, and root-to-tip regression of genetic distance with sampling times was used to determine an approximate age of the root of the phylogenetic trees. The HKY nucleotide substitution model was selected as the best nucleotide substitution model using jmodel test [34]. The models were run with a lognormal uncorrelated relaxed molecular clock and the Bayesian Skygrid population model. To infer transition rates between discrete locations, we used an asymmetric model with Bayesian Stochastic Variable Selection (BSSVS) to identify non-zero rates of transition in the phylogeographic matrix [35]. To reconstruct the evolutionary history, two replicate MCMC chains, with a length of 300 million iterations and sampled every 3000 steps, were performed with computational resources in CIPRES (https://www.phylo.org). Convergences of parameter posteriors were assessed using Tracer 1.6 [36], ensuring effective sample size (ESS) greater than 200 for all parameters. Tree annotator v10.2.2 [37] was used to create maximum clade credibility (MCC) trees, after discarding 10% of iterations as burn-in and using LogCombiner [32] to re-sample to a lower frequency (every 6000 steps) to reduce computational complexity. Fig Tree (v1.4.3) (http://tree.bio.ed.ac.uk/software/figtree/) and ggtree [38] were used to visualize the trees, with branches colored according to the country and the regions of Vietnam. For discrete traits, Spread3 (v0.9.7) [39] was used to calculate the Bayes factor after disregarding 10% burn-in.

#### 2.4.1. Calculation of Adjusted Transition Rates between Agricultural Zones

When BSSVS is used to infer transition rates between locations, BEAST MCMC chain output contains two pieces of information regarding transitions between locations: (a) the rate coefficient, which is the estimated frequency of transmission between two locations, and (b) a 0/1 indicator of whether that transition is included in the model at a particular MCMC iteration. Importantly, when the indicator equals 0, then the rate coefficient is of little meaning, given that it was not included in the model. We processed the raw BEAST output in R (R packages HDInterval, coda, phytools, castor) such that posterior summaries of rate coefficients (mean and 95% highest probability density (HPD) intervals, summarized by Tracer) were based only on MCMC iterations where the indicator equaled 1, which we refer to as the adjusted rate.

Bayes factors were also calculated for all transition rates, with BF > 3 interpreted as statistical support that a rate was non-zero. However, BF should be interpreted with caution as a measure of population connectivity, as high BF tended to be achieved when there was a clear ancestral location for a discrete transition rate, as might be the case for a rare viral dispersal event between different countries. However, the ancestral location may be difficult to determine in situations where there is frequent, bi-directional transmission between locations (yielding high adjusted transition rates, but low BF support on the ancestral location). For this reason, we utilized adjusted transition rates rather than BFs as a proxy for population connectivity in subsequent analyses.

Heat maps and networks were used to visualize viral movement between discrete locations (based on the adjusted rate), created separately for each lineage and the combined analysis. The adjusted rates were used to create the 0/1 phylogeographic matrices (one each for O/ME-SA/Pan Asia, O/SEA/ Mya-98, and combined), using the median value as the threshold to determine 0/1. As the optimal threshold value is unknown, an alternative cut-off value (set at 20% below the median) was additionally investigated. These matrices were used to describe the population connectivity between zones within Vietnam in the space–time regression of reported case counts.

#### 2.4.2. Analysis of Interspecies Transmission

While our model focused on outbreaks in bovines, we conducted a phylodynamic analysis (as described above) using host species as a discrete trait to better understand the role of different host species. The analysis used 240 sequences from Vietnam (160 from cattle and buffaloes, 80 from pigs), and consisted of 7 Cathay, 63 Mya-98, and 170 PanAsia lineage sequences.

### 2.5. Bayesian Space–Time Risk Models

Because the majority of reported outbreaks and sequences were from bovines (71% of sequences, and 72% of reported clinical cases) and based on the results of the phylodynamic analysis of host species, we focused our space–time regression analysis on outbreaks reported in cattle and buffalo. We first calculated standardized incidence ratios (SIRs) per province and year from 2007 to 2017. Bovine population size data were available for the year 2018 only, and we assumed that these numbers were reasonably stable across the time period assessed here. Population size and nationwide FMD reported case counts were used to calculate the expected number of FMD cases per province per year (*e_it_*) if the distribution of FMD cases across space and time was proportional to population size, such that:$$e_{it}\;=P_{it}\;\frac{\sum_{it}^{}Y_{it}}{\sum_{it}P_{it}}$$
where *Pi* is the population of the province *i* in year *t*, and *Y_it_* is the number of FMD cases in province *i* in year *t*. SIRs were calculated as the observed to the expected ratio (*Y_it_*/*e_it_*). SIRs were plotted as choropleth maps for all years.

The observed number of cases per province and year was assumed to follow a Poisson distribution of the form $y_{it}\sim Poisson\;\left(e_{it\;},\theta_{it}\right)$, with $e_{it}$ representing the expected number of FMD cases defined as above, and $\theta_{it}$ representing the yearly relative risk for each province. This relative risk incorporates both spatially structured (spatial correlation amongst connected provinces) and unstructured (i.e., random variation) effects, such that:$$\log\left(\theta_{i\;}\right)=\alpha +\upsilon_{s}+\nu_{s}\;$$
where α is the intercept representing the overall level of risk in the country, *ʋ_s_* is the structured spatial effect, and *ν_s_* is the unstructured random effect for each province. Models of this form are sometimes referred to as BYM2 models [40]. The structured spatial effect incorporates correlations amongst neighboring provinces, conventionally represented by a 0/1 spatial adjacency matrix. The model that utilized inferred viral movement (instead of spatial adjacency) as a proxy for population connectivity mirrored the above space–time model, except that the structured effect was designed to account for correlations amongst provinces connected phylogeographically (*ʋ*_p_) rather than spatially (*ʋ_s_*). The matrix of inferred movement of the virus among the eight agricultural zones inside Vietnam was projected to create a 0/1 matrix for 63 provinces inside Vietnam (i.e., all provinces within the same zone received the same phylogeographic transition rates); conceptually, this accounts for a potential increase in case counts within a given province as a result of viral movement into its agricultural zone. Six phylogeographic matrices were considered (O/ME-SA/Pan Asia, O/SEA/Mya-98, and combined, each with two cut-off values for dichotomization of adjusted rates). We also explored whether model fit could be improved by incorporating both spatial and phylogeographic adjacency matrices simultaneously in the same model or by utilizing a single joint matrix representing connectivity either by spatial or phylogeographic connections.

Several model structures exist to incorporate temporal effects, specifically, time (year) can be considered as a random effect (ω_t_), or a structured effect (γ_t_) in which a random walk is used to account for between-year dependencies (ω_t_ + γ_t_).

The best-fit model structure was selected from amongst these models using DIC. This model was then used to evaluate the contribution of hypothesized spatial risk factors in shaping relative risk. Penalized priors were used for all models in the Bayesian analysis, following previous studies [41].

#### 2.5.1. Incorporation of Spatial Risk Factors as Fixed Effects

To further explain spatiotemporal variation in relative risk, we incorporated factors potentially associated with viral movement as fixed effects into the best-fit spatial and phylogeographic regressions from above. These included two approaches for accounting for transboundary introductions. First, provinces were categorized as to whether they had an international border (0/1). Second, we used the inferred adjusted transition rates from the phylogeographic models to categorize provinces according to whether there was evidence of FMDV introductions from neighboring countries. As before, adjusted transition rates were dichotomized at the median (0/1) to create one dummy variable each for phylogeographic-inferred transition rates from Cambodia, Malaysia, Laos, China, and Thailand into Vietnam provinces. Given that slaughterhouses are terminal points of supply chains and thus may influence animal movements, the presence of a slaughterhouse in a province was also included as a potential 0/1 risk factor in the model. Slaughterhouses considered here are ones that operate under veterinary control and commercial fresh meat establishments registered for export by the national veterinary services (PVS Vietnam, WOAH). Finally, to account for the presence of other hosts that can transmit FMDV [42], goat and pig densities were included as potential risk factors.

Prior to model selection, correlations between variables were checked using Pearson’s correlation coefficients for continuous variables and chi-square tests for categorical variables. In the latter case, an odds ratio of >8 was considered evidence for collinearity [43]. In the case of collinearity, only the variable with the lowest DIC in its respective univariable model was retained for the multivariable analysis. Backward selection was then performed from a full multivariable model by removing the variables with the widest confidence interval that overlapped zero. From among those different models, the simplest model that was <2 ∆DIC from the model with the lowest DIC value was considered the best-fit model.

#### 2.5.2. Prior Sensitivity Analysis and Evaluation of Model Fit

We used non-informative penalized complexity priors, which are applicable for a large class of hierarchical models. The penalized priors aim to incorporate minimal information into the inference procedure, and can account for overdispersion in the base model [44]. A sensitivity analysis was conducted using different priors. We also calculated the correlation between the predicted and observed values (Spearman’s correlation) as a measure of model fit.

The final best-fit model was also evaluated based on posterior predictive *p*-values. Posterior predictive *p*-values are defined as p (y_i_ * ≤ y_i_|y), where y_i_ * is the posterior of the predicted distribution from the model. This is interpreted as an approximation of the proportion of the predicted distribution for y_i_ that is more extreme than the observed value, and values of p (y_i_ * ≤ y_i_|y) near 0 and 1 indicate poor model fit [4]. To better interpret correlations in case counts across different areas, we tabulated bovine case counts per agricultural zone from 2007 to 2017 and created a heatmap showing the Pearson’s correlation in case counts between the eight different agriculture zones across years.

#### 2.5.3. Software for Space–Time Regression

All analyses were performed in the R statistical software (R core team 2020), using packages tidyverse 1.2.123, spdep 0.7–425 [45], dplyr [46], stringr [47], boa [48], viridis [49], ggpubr [50], readr [51], igraph, [52] and ggplot2. For the Bayesian risk models, INLA 19.09.03 [53] was used, and model results were processed with INLAOutputs 19.09.03 [54].

## 3. Results

### 3.1. Phylogeographic Analyses

We conducted a discrete-space phylogeographic analysis of FMDV serotype O in Vietnam, with separate analyses performed for the 229 O/ME-SA/PanAsia sequences and 146 O/SEA/Mya-98 sequences, and for the 400 combined serotype O sequences. Across all three phylogeographic models, the mean substitution rate was 0.0069 substitutions/site/year (95% HPD interval: 0.0056–0.0083) for O/ME-SA/Pan Asia, 0.0051 (0.0038–0.0063) for O/SEA/Mya-98, and 0.0062 (0.0053–0.0071) for the combined serotype O analysis. Here, we present the results from the O/ME-SA/PanAsia phylogeographic model, given that inferred viral movements from this model provided the best-fit to the outbreak risk model (see below), whereas corresponding results for the O/SEA/Mya-98 and combined phylogeographic models are shown in Appendix A. The MCC tree created based on the PanAsia sequences shows the most likely movement of O/ME-SA/PanAsia amongst different regions of Vietnam and surrounding countries (Figure 1). According to the tree, the PanAsia strain moved from Laos and China to Vietnam and Malaysia.

The heatmap depicting adjusted transition rates between countries and agricultural zones is shown in Figure 2A. Of note, there was substantial movement inferred between spatially adjacent zones, as well as more distant zones, such as from the South-Central Coast and Southeast to the Northeast. Transition rates were dichotomized at the median adjusted rate (0.575 for O/ME-SA/Pan Asia, 0.666 for O/SEA/Mya98, and 0.409 for combined sequences), and used to create the 0/1 matrix utilized in the outbreak risk model. A network representation of this matrix is shown in Figure 2B, revealing a combination of local and long-distance connections between zones.

Although the majority of sequences and reported outbreak data came from cattle and buffalo, we performed an additional discrete-trait analysis for host species to further explore the potential role of interspecies transmission from pigs in Vietnam. This model showed little evidence that viruses circulating in pigs were transmitted into bovine populations in Vietnam (Bayes factor = 0.51), though there was evidence of viral transmission from bovines to pigs (Bayes factor = 77,512.2) (Appendix A and Appendix A). This result provided an additional rationale supporting our decision to focus subsequent space–time risk analyses on reported case counts in bovines.

### 3.2. Space–Time Risk Model

The SIR values calculated for 2007 to 2017 used for the Bayesian analysis are shown in Figure 3. Provinces with SIR greater than one can be interpreted as areas with more reported FMD cases than expected given the size of their bovine population. Several different models were tested to select the best model structure for the space–time regression (Table 1), including different combinations of adjacency matrices (spatial, phylogeographic, or both) and temporal effects. The best-fit model incorporated time as both a random walk and random effect and utilized the O/ME-SA/PanAsia phylogeographic matrix, with the threshold for the phylogeographic matrix set at the median adjusted rate (Table 1). This model performed better than the risk models based on O/SEA/Mya-98 or the combined serotype O sequences models. This model also performed better than models that utilized the spatial adjacency matrix alone or phylogeographic and spatial matrices in combination.

The best-fit model structures for the conventional space–time and phylogeographic risk models were used as the base for multivariable models that included additional fixed effects potentially associated with animal movement (i.e., presence of slaughterhouses, presence of an international border, evidence of phylogeographic links with neighboring countries, and host densities). Fixed variables were screened for correlations using Pearson’s correlation coefficients and Chi-Square tests (Appendix A). If two variables were found to be correlated, we excluded the variable that resulted in higher DICs when assessed in univariable models. In the multivariable analyses, inferred virus movement from Cambodia and Malaysia were highly correlated with movement from Thailand, China, and Laos. Virus movement from Cambodia and Malaysia were also highly correlated with each other as well. Therefore, for the space–time risk model, only virus movements from Thailand, China, Laos, and Cambodia were considered in the multivariate analysis, and for the phylogeographic risk model, only viral movements from Thailand, China, Laos, and Malaysia were considered in the multivariate analysis. The best-fit multivariable models are shown in Table 2 and Table 3. Both the spatial and phylogeographic risk models identified slaughterhouses and international borders as associated with risk (95% credible intervals did not overlap 0). In the phylogeographic risk model, additional significant factors included pig density and inferred viral movement from China and Malaysia. Overall, the phylogeographic multivariable risk model had a lower DIC than the space–time model. Several areas in north, south, and central regions (Ha Giang, Cao Bang, BacCan, Lang Son, Ha Tinh, Kon Tum, DakLak, Can Tho) of the country were identified from both best-fit models as high-risk areas through time (Appendix A). A heatmap was used to visualize correlations in case counts between different zones across time (Figure 4). Of note, case counts in geographically distant zones, such as the Northeast and the South-Central Coast, were sometimes more correlated with each other than closer regions (Figure 4).

Results from the prior sensitivity analysis show that results were consistent across different penalized complexity priors (Appendix A). The observed SIR to fitted relative risk values were compared (Spearman’s correlation coefficients = 0.9). For the selected models, the posterior predictive *p*-values indicated good fits (phylogeographic model 19.2 to 79.9, phylogeographic and spatial model 20.3 to 79.2, spatial model 19.5 to 80).

## 4. Discussion

Population connectivity is fundamental to the spread of infectious diseases, and it is important for space–time risk models to account for potential long-distance connections amongst different host meta-populations. In the absence of data on host movement, such long-distance connectivity can be inferred from viral phylogeographic movements that are potentially associated with patterns of host movement in the country. In the current study, we demonstrated that accounting for population connectivity through phylogeographic rate matrices explained more variability in space–time regressions of reported FMD cases than spatial adjacency amongst provinces.

In our analysis, the phylogeographic connectivity network inferred from the sequence data was best able to explain spatial variation in reported clinical case counts, and hence may better capture population connectivity amongst regions than spatial adjacency alone. Our findings are similar to [55], where pathogen sequence data from *Mycobacterium bovis* were utilized to infer patterns of host movement in Cameroon. In that study, the authors had an observed cattle movement network to which the molecular-based network could be compared. They found that the molecular network was a much better approximation of the observed host movements than other methods commonly used in the absence of movement data, such as gravity models [55]. In the United States, patterns of spatial diffusion of Porcine reproductive and respiratory syndrome virus were inferred via discrete-space phylogeographic models, and variability in the rate of sector-to-sector diffusion was shown to be associated with the movement of feeder pigs [56]. Collectively, these studies help validate the use of pathogen molecular data as a proxy for host movement at relatively fine scales.

Because there was uncertainty about how to quantify the phylogeographic data, we tested multiple formulations of the phylogeographic matrix based on different lineages and rate thresholds. We found that the O/ME-SA/Pan Asia phylogeographic matrix provided the best fit in the space–time regression of reported clinical cases. This may be unsurprising given that most detected sequences circulating in Vietnam belonged to this lineage during this period. In addition, we noted correlations in case counts between geographically distant areas, such as the Northeast and the South-Central Coast, suggesting that distant regions may experience synchronized outbreaks (Figure 4). Such correlations would not be captured by spatial adjacency and are better captured by phylogeographic connectivity. Indeed, high adjusted rates were inferred from the Northeast to the South-Central Coast in the phylogeographic analysis (Figure 2A), which is consistent with general patterns of movement of ruminant livestock described for the region [17]. This also aligns well with spatial hotspots of FMD outbreaks previously identified in the Northwest, Northeast, and Red River Delta agroecological zones by a spatial clustering analysis [18].

Previous analysis of the phylogeography of the O/ME-SA/Pan Asia lineage in Vietnam from 2010–2014 showed that FMDV circulated throughout the country, with a special emphasis on the North and South regions [15]. In parallel to that study, we also found that Vietnam was likely to be the origin of viral movements into China, rather than the recipient of viral introductions from China. Although these results could be affected by minimal data availability from China, it is consistent with the general directionality of movement of ruminant livestock from SEA northward into China [17]. This directional movement may contribute to the protective effect of inferred viral movements with China in our risk model, as the risk model accounted for the directionality of viral movements between Vietnam and other countries (Table 3). In contrast to [15], we included older FMDV sequences available from adjacent countries prior to 2010 (Figure 1), showing the introduction of O/ME-SA/Pan Asia into Vietnam from other SEA countries. Recent FMDV sequences from adjacent countries (after 2010) were not available from GenBank, which prevented us from further examining bidirectional transmission patterns in more recent years.

Adding additional fixed effects reduced the DIC of both models, indicating that fixed effects were able to explain some observed variability in both models. Provinces with international borders showed higher relative risk of reported cases, a pattern which held in regressions that used both the spatial and phylogeographic matrices. Even though pig populations are larger than bovine populations (and pig and bovine populations are not highly correlated, ρ = 0.45), pig density had a protective effect on occurrence of FMD outbreaks in bovines. Also, while we expected that the presence of a slaughterhouse in a province would increase the risk of outbreaks, the opposite appeared to be true. This could be a result of the narrow definition of slaughterhouses utilized by OIE performance of veterinary service (PVS) analysis, where we retrieved data for this variable, which may not have captured smaller non-designated abattoirs across the country. If designated slaughterhouses and higher pig-density are located in peri-urban areas where ruminant densities are low, these confounding factors could explain the protective effect of their presence in our models.

Experimental studies have shown that, once infected, pigs transmit the virus more readily to other species compared to bovines [42]. Pigs are also less likely to be infected via the aerosol route [42]. Despite this, our discrete-train analysis of host species suggests that the virus generally moves from bovine populations into pigs in Vietnam, and not in the reverse direction (Appendix A), though this finding should be viewed with caution due to limited sequence availability from pigs (both numerically and geographically). In contrast, using a smaller dataset focusing on a shorter time range, Brito et al. [15] showed that O/ME-SA/Pan Asia moved from pigs to bovines in Vietnam. These contrasting results illustrate how analysis of different serotypes, lineages, or timeframes could yield different results; results of both studies were likely influenced by sampling biases towards ruminants. The conflicting results of these studies highlights the need for more research with representative sampling to better quantify the extent of cross-species transmission in Vietnam. That being said, another recent study also suggested a larger role for ruminants than pigs in the spatial spread of the virus in Vietnam based on husbandry practices in the country [57].

As a result of the over-representation of bovines in both the outbreak and sequence datasets, our space–time regressions describe patterns of O/ME-SA/PanAsia within bovine populations and may not capture what may be occurring in pig populations. An additional limitation is that, because sequence data were not available from all provinces, agricultural zones were used to infer patterns of viral movement, which was then interpolated to the province level for the space–time regressions. Therefore, the influence of phylogeographic connections on case counts within a given province can be interpreted as an increased risk of outbreaks at the province-level as a consequence of viral introductions into (and subsequent circulation within) the broader agricultural zone. Observed patterns may change with the inclusion of additional, more broadly representative sequence data from other host species, provinces, or viral lineages, particularly if the dominant viral lineage circulating in the country changes.

Another limitation of our study was that we used the number of animals (reported case counts) as our outcome as opposed to the number of reported outbreaks. However, case counts and outbreak counts both have limitations; case counts may allow a single large outbreak to have too much influence on the results, whereas outbreak counts treat large and small outbreaks equally. We chose to use reported case counts, as it is unclear what criteria are used to classify a large group of cases into one large outbreak as opposed to several smaller outbreaks. Additionally, it is likely that reported numbers are an underestimate of true disease incidence. For example, there was a high reported number of cases from the years 2011 and 2012 in Cao Bang province. This could be due to a fast-spreading FMD outbreak or because of increased surveillance in the area during the period. This could have impacted our model, for example, by flattening the impact of the temporal effects in other years with the result of identifying similar predicted risk through time. Despite these limitations, this study provides compatible results with other studies [8,13,15,18,25] regarding animal and FMDV movement in Vietnam.

## 5. Conclusions

In this study, we investigated whether the effects of population connectivity in space–time regressions of FMD case counts in Vietnam were better captured by spatial adjacency or by inferences from phylogeographic analyses. Because phylogeography can infer historical patterns of FMDV movement, we found that phylogeographic connectivity was better able to explain variability in case counts than spatial adjacency. While the role of long-distance population connectivity in shaping patterns of disease spread is well understood for highly mobile host populations, our approach provides a means to integrate such information into Bayesian space–time models of outbreak risk when data on host mobility is not readily available. More generally, outbreak risk in Vietnam was correlated with proximity to international borders and inferred connectivity to regions within and beyond Vietnam, further highlighting the importance of animal movements in shaping FMD risk. Such an approach could be adapted for risk modeling in other host–pathogen systems where pathogen sequence data is more readily available than host movement data. Additionally, these findings could be used to enhance endemic countries’ progress through the stages of the Progressive Control Program for FMD (PCP-FMD), ultimately contributing to regional control of FMD.

## Figures and Tables

**Figure 1 viruses-15-00388-f001:**
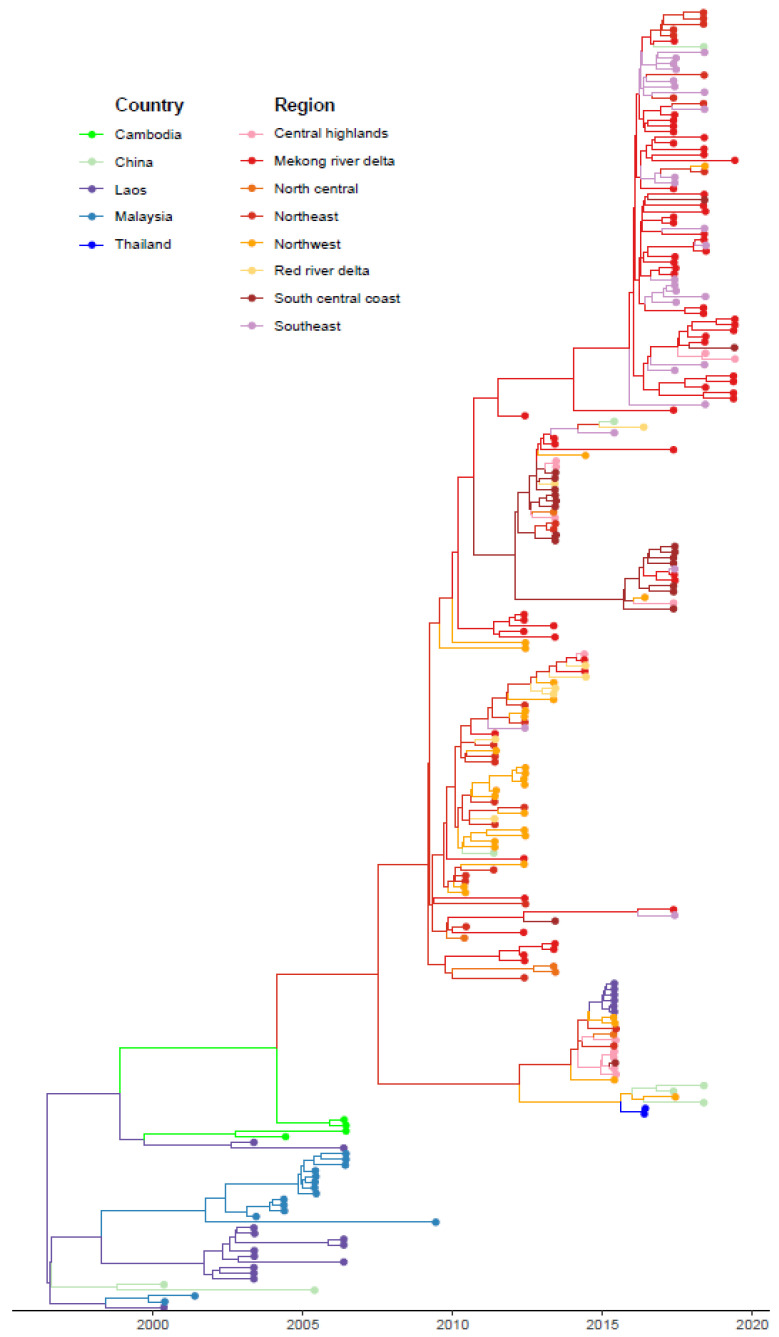
Maximum clade credibility tree for FMDV O/ME-SA/Pan Asia from agricultural zones of Vietnam and surrounding countries of China, Laos, Malaysia, Thailand. Nodes and branches of the tree are colored by location.

**Figure 2 viruses-15-00388-f002:**
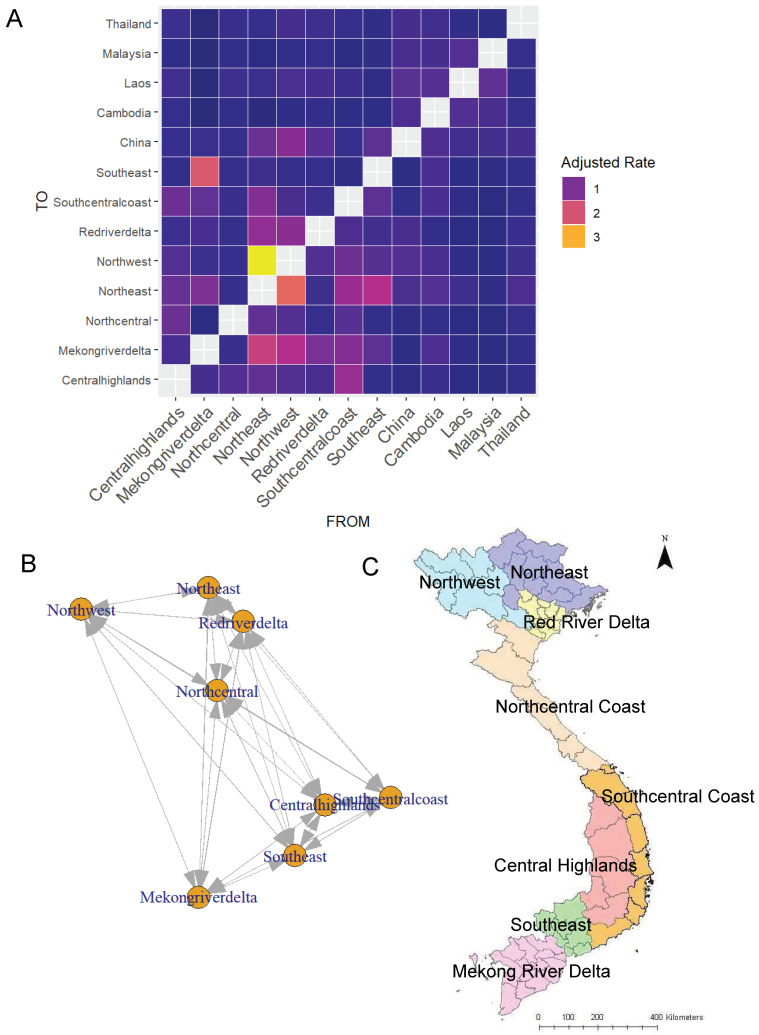
(**A**) The adjusted rate matrix for O/ME-SA/Pan Asia showing the virus movement between the different agricultural zones in Vietnam and the adjacent countries. The color gradient of the heat map indicates the adjusted rates, and colors closer to yellow show higher adjusted rates/higher movement compared to purple, which indicates lower adjusted rates/less movement. (**B**) Network of phylogeographically connected zones within Vietnam showing which regions are highly connected. (**C**) Map of agricultural zones in Vietnam that was used to create the network and the adjusted rate matrix.

**Figure 3 viruses-15-00388-f003:**
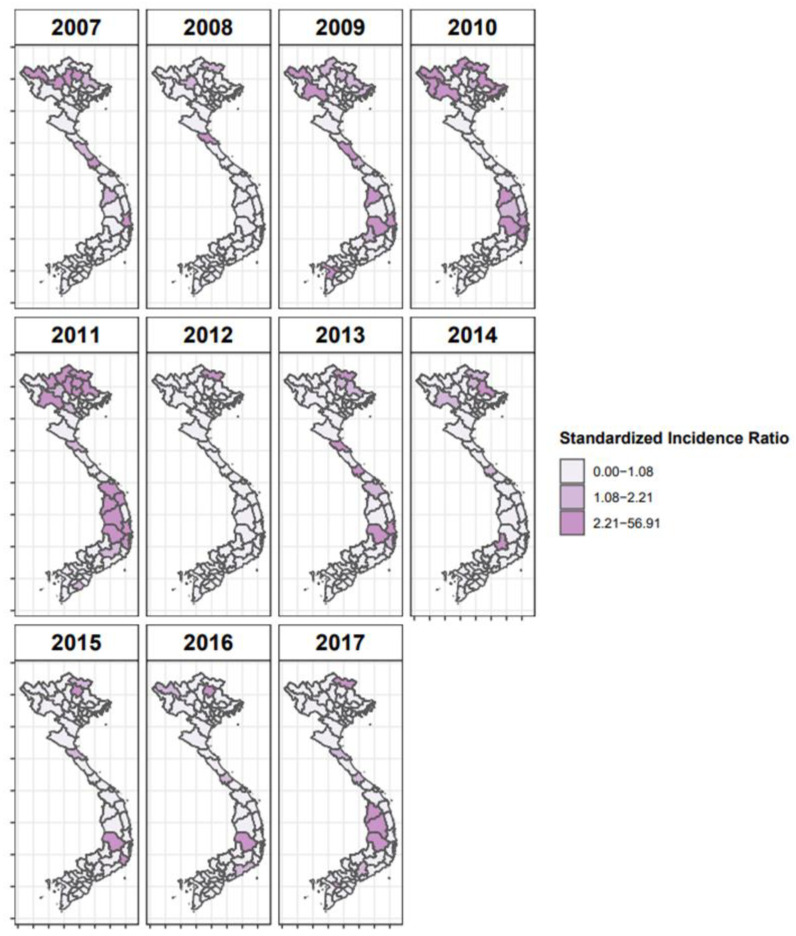
Map of standardized incidence ratios for years 2007–2017 in Vietnam considering the reported outbreak numbers of cattle and buffaloes. Darker areas indicate high-risk provinces during the study period.

**Figure 4 viruses-15-00388-f004:**
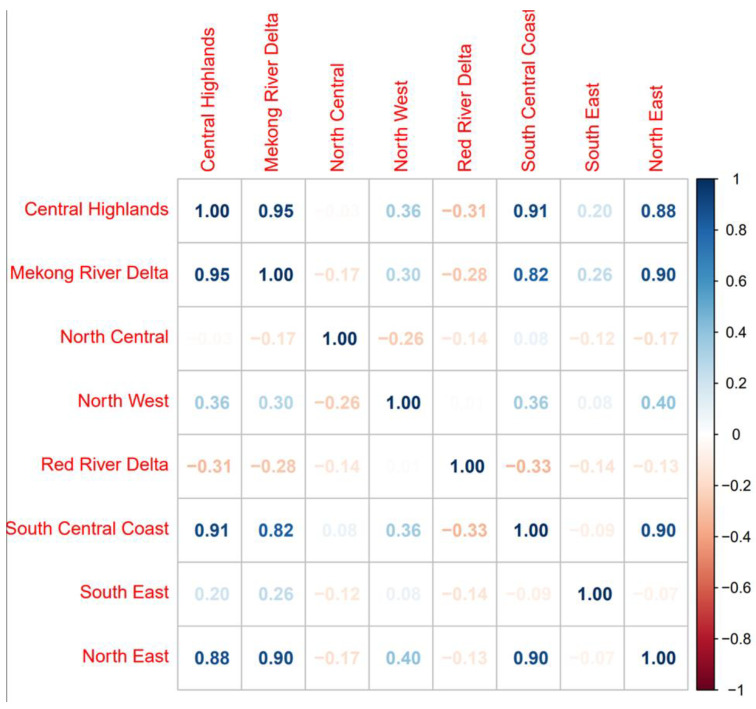
Spearman’s correlation matrix of reported case counts across agricultural zones from 2007–2017. Higher correlation values are shown in darker blue colors and negative correlation values are shown in red color according to the scale shown on the right side.

**Table 1 viruses-15-00388-t001:** Comparison of different model structures utilized for Bayesian space–time regressions of reported case counts. Phylogeographic matrices were tested with two different cutoffs (at median and 80% of median) to dichotomize the phylogeographic adjusted rate matrices. The best-fit structure is marked in bold.

Model Group	Temporal Effect	Lineage	Cutoff	DIC
Spatial	Random effect only	NA	NA	113457.7
	Random effect and random walk	NA	NA	112329.1
Phylogeographic	Random effect and random walk	Mya 98	0.67	110658.8
	Random effect and random walk	Mya 98	0.55	110658.8
	**Random effect and random walk**	**Pan Asia**	**0.60**	**110607.7**
	Random effect and random walk	Pan Asia	0.47	110623.9
	Random effect and random walk	Total sequences	0.41	110622.9
	Random effect and random walk	Total sequences	0.37	110622.9
Phylogeographic and spatial	Random effect and random walk	Mya 98	0.67	112450.0
	Random effect and random walk	Mya 98	0.55	112450.0
	Random effect and random walk	PanAsia	0.60	112710.0
	Random effect and random walk	PanAsia	0.47	112464.6
	Random effect and random walk	Total sequences	0.41	112631.6
	Random effect and random walk	Total sequences	0.37	112631.6
Phylogeographic and spatial (joint)	Random effect and random walk	Pan Asia	NA	110611.3

**Table 2 viruses-15-00388-t002:** Results from the final Bayesian space–time regression model based on the spatial adjacency matrix (DIC 112300).

Fixed Effect	Coefficient (95% Credible Interval)
Intercept	−2.56 (−3.49, −1.65)
Presence of slaughterhouse (ref: no slaughterhouse)	−4.73 (−7.09, −2.55)
International border (ref: no international border)	0.82 (0.18, 1.48)

**Table 3 viruses-15-00388-t003:** Results from the final Bayesian space–time regression based on the phylogeographic matrix (DIC 109366.1).

Fixed Effect	Coefficient 95% Credible Interval
Intercept	−1.35 (−1.53, −1.35)
Pig density	−0.58 (−0.63, −0.53)
Presence of Slaughterhouse (ref: no slaughterhouse)	−2.70 (−2.88, −2.51)
International border (ref: no international border)	0.38 (0.34, 0.42)
Inferred viral movement from China	−0.57 (−0.68, −0.49)
Inferred viral movement from Malaysia	0.79 (0.64, 0.94)

## Data Availability

All data used in this study (FMD outbreak information and sequence data) are already published and are mentioned with references [18,25].

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
