# Peer review of "Phylogeography as a Proxy for Population Connectivity for Spatial Modeling of Foot-and-Mouth Disease Outbreaks in Vietnam"

_viruses, 2023, doi:10.3390/v15020388_

Round 1
Reviewer 1 Report
Manuscript entitled as : Phylogeography as a proxy for population connectivity for spatial modeling of foot-and-mouth disease outbreaks in Vietnam authored by Umanga Gunasekara et al is a very informative and interesting. This manuscript will add use full information regarding the phylogeography as a proxy for population connectivity and its link to disease transmission. This is help full specially in diseases like Foot-and-mouth disease control having transboundary nature.
Line 42-43: reference need to be cited
Line 45-46: The name should be italicized “Aphthovirus” and “Picornaviridae” according to https://ictv.global/faq/names
Line 80: Population description
Here it is important to describe the husbandry system practiced in the country. This has impact on the disease transmission between animals(same species, interspecies), farms, zones and even between provinces. In addition, the livestock marketing system in the country and the surrounding area need to be explained as this also influences disease transmission
Figure 2A: what are the additional color coding in the heat map(rate 1,2, and 3 are indicated with different colors) but colors out of this are visible in the heat map. Can the author explain this in the text or the legend.
Figure 2B: Line 347-348 explains a network representation of this matrix revealing a combination of local and long-distance connections between zones. Can the author add a network representation showing the neighboring countries.
Line 364-368: This need to be connected with the husbandry system practiced in the country and the analysis should be done for other species like buffalo, sheep and goat.
Author Response
We would like to thank the reviewer for their valuable suggestions and following changes were made accordingly.
Manuscript entitled as : Phylogeography as a proxy for population connectivity for spatial modeling of foot-and-mouth disease outbreaks in Vietnam authored by Umanga Gunasekara et al is a very informative and interesting. This manuscript will add use full information regarding the phylogeography as a proxy for population connectivity and its link to disease transmission. This is help full specially in diseases like Foot-and-mouth disease control having transboundary nature.
Line 42-43: reference need to be cited
References were included
Fountain-Jones NM, Kraberger S, Gagne RB, Trumbo DR, Salerno PE, Chris Funk W, Crooks K, Biek R, Alldredge M, Logan K, Baele G, Dellicour S, Ernest HB, VandeWoude S, Carver S, Craft ME. Host relatedness and landscape connectivity shape pathogen spread in the puma, a large secretive carnivore. Commun Biol. 2021 Jan 4;4(1):12. doi: 10.1038/s42003-020-01548-2. PMID: 33398025; PMCID: PMC7782801.
Simon Dellicour, Rebecca Rose, Nuno Rodrigues Faria, Luiz Fernando Pereira Vieira, Hervé Bourhy, Marius Gilbert, Philippe Lemey, Oliver G. Pybus, Using Viral Gene Sequences to Compare and Explain the Heterogeneous Spatial Dynamics of Virus Epidemics, Molecular Biology and Evolution, Volume 34, Issue 10, October 2017, Pages 2563–2571, https://doi.org/10.1093/molbev/msx176.
Line 45-46: The name should be italicized “Aphthovirus” and “Picornaviridae” according to https://ictv.global/faq/names
This was made italic
Line 80: Population description
Here it is important to describe the husbandry system practiced in the country. This has impact on the disease transmission between animals (same species, interspecies), farms, zones and even between provinces. In addition, the livestock marketing system in the country and the surrounding area need to be explained as this also influences disease transmission.
Population description paragraph has been modified as follows as per the reviewers suggestions, line 84
“The livestock market system in SEA is complex. Both livestock and meat products are transported across borders following a system where supply meets the highest demand. There is evidence that livestock move across Vietnam to Thailand via surrounding countries (Di Nardo 2011) and more recent documentation of livestock movement towards China from Lao PDR, Thailand and Vietnam (Polly et al., 2019). Vietnam is a hub for animal movements in the SEA region and cannot be considered as a separate entity.
The country is divided into 63 provinces grouped in eight major agriculture zones, referred to as Northwest, Northeast, Red River Delta, North Central Coast, South Central Coast, Central Highlands, Southeast, and Mekong River Delta. Vietnam has had an FMD control program in place since 2006 (National Program for Prevention and Control of Foot-and-Mouth Disease), and at present, Vietnam is in stage 3 of the OIE/FAO progressive control pathway (PCP). Biannual vaccination of cattle and buffalo is conducted free-of-charge in border provinces and for a fee in other areas of the country [14]. Most (about 85%) of livestock farms in Vietnam are small-scale farms [15]. Pig production supplies 77% of the total meat production in Vietnam. Pig production system has been moving from small-scale backyard systems to large-scale commercial farming system (Dinh T.X 2017). There is evidence that pigs from the small-scale farmers move across different agriculture zones for slaughter and production purposes, especially zones in northern part of the country are highly connected for trading purposes (Baudon et al., 2017). Cattle and buffalo production stand at 18% and 4%, respectively. However, cattle and buffalo farms are more evenly distributed across the country compared to the pig farms concentrated in the northern part of the country (FAOSTAT 2020). Most cattle and buffaloes are reared under extensive management systems for various purposes such as trade, meat production, and draught ( Di Nardo 2011,Dinh T.X 2017). “
Figure 2A: what are the additional color coding in the heat map(rate 1,2, and 3 are indicated with different colors) but colors out of this are visible in the heat map. Can the author explain this in the text or the legend.
A description was included in the legend; line 375-378
“The color gradient of the heat map indicates the adjusted rates, colors closer to yellow shows higher adjusted rates/higher movement compared to purple, which indicate lower adjusted rates/less movement.”
Figure 2B: Line 347-348 explains a network representation of this matrix revealing a combination of local and long-distance connections between zones. Can the author add a network representation showing the neighboring countries.
This figure could be made, but we don’t feel it would be very informative since there would be many connections that would obscure the pattern of connections within Vietnam, which is the focus of the outbreak risk model. Transboundary connections are not considered dynamically in the risk regression.
Line 364-368: This need to be connected with the husbandry system practiced in the country and the analysis should be done for other species like buffalo, sheep and goat.
We have added the below text to the discussion, referring readers to another study on the association husbandry practices in Vietnam in line 534-536
“That being said, another recent study also suggested a larger role of ruminants than pigs in the spatial spread of the virus in Vietnam based on husbandry practices in Vietnam.”
Do, H.; Nguyen, H.-T.-M.; Van Ha, P.; Kompas, T.; Van, K.D.; Chu, L. Estimating the Transmission Parameters of Foot-and-Mouth Disease in Vietnam: A Spatial-Dynamic Kernel-Based Model with Outbreak and Host Data. Preventive Veterinary Medicine 2022, 208, 105773, doi:10.1016/j.prevetmed.2022.105773.
The analysis was done with “bovines” as the species group, which included both cattle and buffalo (we have revised the text to make this clear. While theoretically one could consider buffalo and cattle separately, GenBank data often just lists the host species as “bovine,” so its not always clear whether they come from cattle or buffalo. So, buffaloes were indeed included in the analysis (supplementary table S1 and supplementary figure S3). There was not enough sequence data available for goats and sheep from our project or GenBank.

Reviewer 2 Report
The authors investigate which of two approaches, spatial adjacency or inferences from phylogeographic analysis, better captures population connectivity in spatiotemporal regression of case counts. In order to compare these two methods, the authors used foot-and-mouth disease virus (FMDV) outbreak data from various parts of Vietnam as an example. It was found that interpreting viral motion through phylogeographic analysis better represented population connectivity than spatial adjacency in spatiotemporal risk models. This approach may be useful in designing surveillance activities in countries where mobility data are lacking.
Overall the referee has no critical comments regarding of performance of the investigations. The analysis procedures are OK and the results are adequate for publication.
Author Response
We thank you for reviewer's valuable comments. We have made some additional changes in the manuscript to provide introduction. There were no specific comments to address.
